# The Impact of Heart Rate Variability Biofeedback on Anxiety Reduction and Batting Performance Enhancement in Taiwan University Baseball Players

**DOI:** 10.3390/jfmk10010065

**Published:** 2025-02-13

**Authors:** Yun-Ting Su, Po-Hsun Huang, Tzu-Chien Hsiao

**Affiliations:** 1Institute of Biomedical Engineering, National Yang Ming Chiao Tung University, Hsinchu 30010, Taiwan; suyunting.cs04@g2.nctu.edu.tw; 2Department of Computer Science, College of Computer Science, National Yang Ming Chiao Tung University, Hsinchu 30010, Taiwan; pohsun@nycu.edu.tw; 3Institute of Computer Science and Engineering, College of Computer Science, National Yang Ming Chiao Tung University, Hsinchu 30010, Taiwan

**Keywords:** baseball, HRV biofeedback, state anxiety, batting performance

## Abstract

**Background:** A commonly observed phenomenon is that although the players in a baseball team have received the same training content and volume, their batting performance is quite different. As no optimal solution exists for this problem at present, this study attempted to explore the potential of heart rate variability biofeedback (HRVB) to reduce anxiety and improve batting performance in university baseball players. **Materials and Methods:** A total of 18 college baseball players were randomly divided into an experimental group and a control group. Both groups answered questionnaires and had their physiological signals and batting performance measured on the first and last days of the experiment (i.e., days 0 and 10, respectively). Only the experimental group received HRVB training between the first and last days (10 days in total). **Results:** The results showed that before training, no significant differences were found in physiological, psychological, or performance parameters between the two groups. Compared to the control group, following HRVB training, the experimental group showed a notable decrease in cognitive anxiety (before HRVB: 23.56 ± 4.07; after HRVB: 20.11 ± 4.78; *p* < 0.05) and their batting performance improved significantly (batting score increased from 9.8 ± 11.7 to 19.8 ± 12.0 after HRVB; *p* < 0.05). **Conclusions:** This study validated that the use of HRVB can help to improve batting performance and reduce anxiety in college baseball players. Therefore, HRVB can be applied before competition matches, helping the players to perform better.

## 1. Introduction

For a baseball player to be a good batter, hitting and timing mechanisms, as well as a good mental state, are required during batting. Proper hitting mechanisms, including centered gravity, arm positioning, and length of stride, can help the batter to produce a coordinated swing. Timing mechanisms focus on whether and when to swing at the pitched ball, which is correlated with the batter’s visual search strategy and decision-making ability. Mental state during batting is also an important factor affecting batting performance. Empirical and experiential evidence suggest that anxiety is a frequent occurrence in the game, which can have an extremely detrimental effect on athletic performance [1]. Due to the pressure imposed by the game, a batter’s anxiety can unconsciously trigger the activation of their autonomic nervous system (ANS). The ANS comprises sympathetic and parasympathetic nervous systems, which have antagonistic effects on the regulation of bodily functions. ANS responses that are related to emotions include cardiovascular, electrodermal, and respiratory responses [2].

Heart rate variability (HRV) is a common index used to investigate the function of the ANS. The Task Force of the European Society of Cardiology and the North America Society of Pacing and Electrophysiology have developed definitions for HRV terms and specified standard methods of measurement. Typical HRV indices in frequency domain analysis include very low frequency (0.003–0.04 Hz, VLF), low frequency (0.04–0.15 Hz, LF), and high frequency (0.15–0.4 Hz, HF) [3]. Respiratory sinus arrhythmia (RSA) is a physiological phenomenon reflecting respiratory–circulatory interactions [4], where the heart rate (HR) fluctuates in phase with breathing. Heart rate variability biofeedback (HRVB) is a training method that provides a real-time beat-by-beat HR waveform, guiding the subject to breathe at a target frequency, thus maximizing the amplitude of RSA for effects such as increased alveolar gas exchange, mediating homeostasis in the body, and promoting relaxation. HRVB was developed in the 1980s as an intervention for various disorders and performance enhancement [5]. Its theoretical foundation lies in the concept that periodically increasing cardiac period oscillations at specific frequencies can exercise and strengthen baroreceptor reflexes, thereby modulating blood pressure. This particular frequency is referred as resonant frequency (RF), which is centered at 0.1 Hz [6].

Since 2003, several studies have investigated the effects of HRVB training on athletic performance [7,8,9,10,11,12]. While subjects from various sports have demonstrated improved athletic performance following HRVB training, the training protocols varied considerably across studies. The protocol proposed by Lehrer et al. [13] in 2000, involving ten weeks of self-practiced HRVB twice daily, has been employed in two studies. Lagos et al. observed significant improvements in HRV, LF, and HF in golfers who self-practiced twice daily [9]. Similarly, Choudhary et al. (2016) reported a significant decrease in HR, breathing rate (BR), and skin conductance, along with a significant increase in LF/HF, among long-distance runners [11]. In 2003, Strack recruited baseball players and shortened the training period to six weeks with twice-daily self-practice, demonstrating an increase in normalized LF power (nLF) after HRVB [7]. Paul and Garg (2012) recruited basketball players and further shortened the protocol to 10 consecutive days, finding significant increases in total HRV power and LF [10]. Notably, Jian et al. (2019) observed no significant changes in psychophysiological parameters or performance in golfers who received a single ten-minute session of HRVB [12]. Furthermore, the psychological outcomes are inconsistent across studies, with athletes in some sports experiencing reduced anxiety after HRVB, while baseball batters showed no significant change.

Based on previous studies [7,10] and considering the concentrated nature of university baseball seasons, this study adopted the training protocol proposed by [10] while incorporating the HRVB procedures outlined in [7]. In light of the discrepant psychological findings reported in [7], this study aimed to further investigate the effects of HRVB training on anxiety reduction and batting performance in baseball players. Therefore, two hypotheses were formed: (1) HRVB training will reduce anxiety in baseball batters; and (2) HRVB training will improve batting performance in baseball batters. In the following sections, the experimental procedures and analysis methods are detailed, the results of the conducted experiment are presented, and the effects of HRVB from various perspectives are discussed. Finally, a conclusion is given.

## 2. Materials and Methods

### 2.1. Participants Recruitment

The investigation was approved by the campus Institutional Review Board (approval NCTU-REC-108-029F), and 21 male baseball players ranging in age from 19 to 29 years (21.2 ± 2.6 years) were recruited from the college team of National Yang Ming Chiao Tung University (NYCU, formerly known as National Chiao Tung University), Taiwan. All participants provided informed consent. Participants were excluded if they met any of the following criteria: injury, illness, diagnosis of cardiovascular disease or severe asthma, or experience with any type of biofeedback training. Due to the impact of the COVID-19 epidemic, three participants withdrew from the training stage for reasons relating to graduation and coursework. Participants were randomly and equally divided into an experimental group (20.9 ± 3.1 years) and a control group (21.4 ± 2.0 years).

### 2.2. Experimental Procedure

The experiment was conducted in August 2021. The experimental procedure is shown in Figure 1, and all participants completed a demographic questionnaire in the first meeting. The procedures of trial 1 and trial 2 were the same. First, participants were requested to complete questionnaires that included the Competitive State Anxiety Inventory 2 (CSAI-2), which is usually used to measure levels of cognitive anxiety, somatic anxiety, and self-confidence, and the Coping Self-Efficacy Scale (CSES), which is used to measure perceived self-efficacy for coping with challenges or threats. After filling out the questionnaires, physiological signals were acquired via electrocardiogram (ECG) and respiratory inductance plethysmography (RIP). A stress profile, including baseline (4 min), stressor (2 min), and recovery (4 min) phases, was conducted for physiological signal measurement. In the baseline phase, the participant was required to sit comfortably on a chair, while in the stressor phase, the participant was given a math task (“serials seven”) requiring them to subtract 7 from 994. In order to induce stress, the participant had to write down the answer as quickly as possible without making mistakes. In the recovery phase, no other instruction was given.

The final part was measurement of batting performance. A pitching machine was used to pitch at 120 km/h, with each participant given 5 warm-up pitches and 20 regular pitches, and a pre-determined point system designed by Strack [7] was used to score and record each ball hit (Figure 2). Additionally, a swing analyzer (Baseball Swing Analyzer, Blast Motion, Inc., San Marcos, CA, USA) was mounted on the end of baseball bat, in order to record the bat speed (miles per hour, MPH), peak hand speed (MPH), and other parameters.

The training protocols were similar between the two groups; however, the experimental group received 20 min of HRVB per day for 10 days. Both groups had 40 balls of batting practice every day.

### 2.3. Questionnaires

Participants completed a battery of questionnaires, including a demographic questionnaire (assessing gender and age), the CSAI-2, and the CSES. The complete set of questionnaires administered to participants is available in the Appendix A.

The CSAI-2, developed by Martens in 1990, assesses athletes’ anxiety levels across three dimensions: cognitive anxiety, somatic anxiety, and self-confidence [14]. This inventory comprises 27 items and employs a 4-point Likert scale, where respondents rate how they feel at the moment using the options “not at all”, “somewhat”, “moderately so”, and “very much so”.

The CSES measures an individual’s confidence in performing coping behaviors when facing life challenges [15]. This scale consists of 26 items, each rated on an 11-point scale ranging from 0 (“cannot do at all”) to 10 (“certainly can do”), reflecting the individual’s perceived confidence under pressure. An additional 27th item, identical to the 6th item, was included in this study to ensure participants’ attentiveness while completing the questionnaire.

### 2.4. HRVB Training

Participants were recruited from a university where the game season may be a specific month or week of the semester. Therefore, the present study followed the training protocol proposed by Paul and Garg [10], in which participants received 10 consecutive days of HRVB. The training interface was constructed using the LabVIEW (ver 19) programming system, and a pacing stimulus was provided for the training in order to guide the participants to breathe at the target frequencies.

The main purpose of the first day of HRVB was to determine the RF of the participant. The participant was guided to breathe following the pacing stimulus for three minutes at each of the frequencies between 4 and 7 breaths/minute (6.5, 6, 5.5, 5, 4.5 breaths/minute), and their ECG and RIP signals were recorded. The RF was estimated from the best convergence of the following conditions: (1) HR in phase with breathing; (2) the highest peak-through amplitude; and (3) the maximum LF amplitude peak on the spectral graph in ms^2^/Hz [16].

In days 2 to 10 of HRVB training, participants were guided to practice breathing at RF following three instructions: (1) breathe with longer exhalation than inhalation; (2) breathe in through the nose with mouth closed and breathe out through pursed lips; and (3) abdominal breathing, abdomen moving out when inhaling and moving back in when exhaling. After a few minutes of practice, the pacing stimulus was shut off, and the participant was guided to breathe following the HR curves on the training interface.

### 2.5. Data Processing for Physiological Signal

The signal processing flow of the ECG signals is shown in Figure 3. ECG signals were used as input to the LabVIEW programming system, which detected the R-peak and calculated the R-R intervals (RRIs). Analysis of RRIs was divided into time domain and frequency domain, where time domain analysis was conducted to calculate the reciprocal of RRIs and obtain the mean and standard deviation of the HR. Spectral analysis of HRV was carried out using a fast Fourier transform (FFT), after which the HRV indices were calculated.

The present study used the LabVIEW programming system to process RIP signals [17], and the processing flow is shown in Figure 4. First, a digital median filter with value 2 for left and right ranks was used to reduce noise from the abdominal wall movement (AWM) signal. Second, the signals were decomposed into several intrinsic mode functions (IMFs) using complementary ensemble empirical mode decomposition (CEEMD) [18]. The IMF with the highest energy density was chosen as the dominant component, and the BR was calculated.

### 2.6. Statistical Analysis

The Statistical Product and Service Solutions (SPSS 22) software was used for statistical analysis. Descriptive statistics, including the mean (μ), standard deviation (std), and median (Md), were calculated for the phenotype data, such as age, years of experience playing baseball, height, weight, and body mass index (BMI).

Non-parametric statistics were used to describe the basic features of the acquired data due to the small sample. The Mann–Whitney U-test was performed to test the differences between control and experimental groups, and the Wilcoxon signed-rank test was used to observe changes after HRVB. In addition, a chi-squared test was used to determine the relationship between batting distances between the two groups in both trials.

G*Power (version 3.1.9.7) was used to calculate power (1-β) and effect size based on the significance level (α). Since G*Power calculates Cohen’s *d* as the effect size, a metric that may not be optimal for non-parametric statistical methods, we derived the rank-biserial correlation *r* from Cohen’s *d* using the formula $r=\frac{d}{\sqrt{d^{2}+4}}$ [19].

## 3. Results

### 3.1. Participants

Basic information of the enrolled subjects is shown in Table 1. All participants were male, and there was no significant difference between the control and experimental groups in any characteristic.

### 3.2. Statistical Analysis and Descriptive Statistics

The descriptive statistics included changes in physiological parameters, psychological parameters, and batting performance. Values are expressed as means ± standard deviation. The statistical analyses are reported in terms of *p*-values, statistical power, and effect sizes for pairs exhibiting significant differences.

The descriptive statistics for HR and BR values are shown in Table 2. No significant differences in HR values were found between groups and trials, whether in baseline, stressor, or recovery phases. The experimental group presented a significant decrease in BR values in the stressor phase (*p* = 0.028, power = 0.79, effect size = 0.52) following 10 consecutive days of HRVB. In contrast, there was no significant change in the control group.

To gain further insight into the activity of the ANS, frequency domain parameters, including the nLF, normalized HF (nHF), and LF/HF, were calculated, and the results are shown in Table 3. Before training, there were no significant differences between the two groups. However, the experimental group presented a significant increase in nLF in the baseline phase after HRVB (*p* = 0.038, power = 0.29, effect size = 0.29). Moreover, in the baseline and stressor phases of trial 2, significant differences were found in the nLF (*p* = 0.002 and 0.024; power = 0.97 and 0.44; effect size = 0.80 and 0.43, respectively) and LF/HF (*p* = 0.002 and 0.024; power = 0.98 and 0.46; effect size = 0.81 and 0.44, respectively) between the two groups.

To exclude the influence of individual differences, the variations for each parameter were calculated, and the results are shown in Table 4 and Table 5. No significant differences were found in △HR or △BR values. Whether in baseline, stressor, or recovery phases, △nLF (*p* = 0.047, 0.031, and 0.031; power = 0.56, 0.31, and 0.52; effect size = 0.42, 0.33, and 0.44, respectively) and △nHF (*p* = 0.047, 0.031, and 0.031; power = 0.56, 0.31, and 0.52; effect size = 0.42, 0.33, and 0.44, respectively) presented significant differences between the control and experimental groups. Furthermore, a significant difference was found in △LF/HF (*p* = 0.038, power = 0.39, effect size = 0.36) values for the baseline phase.

The descriptive statistics of each questionnaire are shown in Table 6. In trial 1, there were no significant differences in cognitive anxiety, somatic anxiety, self-confidence, or CSES between the two groups. Following the HRVB, cognitive anxiety was significantly reduced in the experimental group (*p* = 0.036, power = 0.34, effect size = 0.32) compared with no change in the control group. No significant differences were found in the CSES score before and after training between the two groups.

The descriptive statistics for batting performance are shown in Table 7. The sum of scores for 20 balls was considered the batting score, and the infield ratio (%) was defined as 20 divided by the number of balls that had been hit infield. Before HRVB, there were no significant differences in batting score or infield ratio between the groups, whereas the experimental group showed significant increases in batting score (*p* = 0.012, power = 0.63, effect size = 0.52) and infield ratio (*p* = 0.017, power = 0.63, effect size = 0.50) after training. The swing section analyzer parameters indicated no significant differences between the two groups before HRVB, while there were significant differences in bat speed (*p* = 0.047, power = 0.59, effect size = 0.43) and peak hand speed (*p* = 0.038, power = 0.58, effect size = 0.45) between the two groups after HRVB.

The pre-determined scoring system (Figure 2) was further divided into four categories, where the first category was 0 points; the second and third categories were 1–3 points and 4–6 points, respectively; and the fourth category was 7 points. The total number of balls in each category was counted; however, as no participant scored 7 in either of the trials, all balls fell into category I, II, or III. The results are shown in Figure 5. The data in Table 8 were used to construct contingency tables for the chi-squared test, examining the relationship between categories II and III across the two groups in the two trials. In both trials, the number of hit balls was similar for the two groups (*p* = 0.950, X^2^ = 0.004), where in trial 1, the number of balls in categories II and III for the two groups was similar (*p* = 0.441, power = 1.00, X^2^ = 0.59); however, a significant difference was found for balls in categories II and III between the two groups in trial 2 (*p* < 0.001, X^2^ = 17.47).

## 4. Discussion

One of the goals of the present study was to validate the effects of HRVB on anxiety reduction, and the results of this study indicated that the experimental group exhibited a considerable reduction in cognitive anxiety after HRVB when compared to the control group, which presented no change. These findings may be attributed to successful mastery of psychophysiological control and the participants reaching a deeper level of relaxation. The other goal was to examine whether HRVB can help in terms of batting performance improvement. The results also showed that the experimental group had better batting scores in trial 2 compared with those in trial 1, while the control group did not show a significant difference between the two trials. Additionally, Figure 5 indicates that compared to the control group, there was a shift in batting scores from category II to category III in the experimental group, meaning that the hit distance was greater following the HRVB training. This finding suggests that batters may have better quality of contact after HRVB. Regarding the ANS assessment, nLF exhibited a significant increase at baseline in trial 2 within the experimental group. While no significant differences in nLF were observed between trials 1 and 2 during the stressor or recovery phases in either group, the average nLF demonstrated an increase in the experimental group and a decrease in the control group. These findings suggest that HRVB may have a stimulatory effect on the sympathetic nervous system, potentially enhancing concentration and thereby improving batting performance. Lehrer et al. proposed that HRVB helps to restore autonomic balance and improve autonomic control [20]. Previous studies have also shown that HRVB is effective in regulating physiological states, with significant changes in ANS-related parameters following 10 days to 10 weeks of HRVB training. This study involved 10 consecutive days of HRVB training, and the testing results indicated that nLF in the baseline phase was increased significantly. Furthermore, significant differences were found in nLF and LF/HF between the two groups after HRVB. The observed increase in nLF may be attributed to the resonance effect of the cardiovascular system [9].

Previous studies have shown that HRVB is effective in improving athletic performance in different sports. Similarly to this study, Strack recruited baseball players to practice HRVB and explored its effects on performance. The obtained results indicated that the biofeedback group improved their batting performance significantly following six weeks of HRVB. In the present study, even though the training duration was shortened to 10 consecutive days, the results indicated that batting score and infield ratio improved significantly in the experimental group. Additionally, significant differences in bat speed and peak hand speed were observed between the two groups in trial 2. It can be inferred from the results that despite shortening the training duration, HRVB is still effective in improving batting performance.

The limitations of the present study are as follows. (1) Although 10 consecutive days of HRVB was conducted, the training effects under other types of training remain unknown. (2) The sample was too small, so the statistical power might be insufficient. A larger sample is required to more definitively determine the impact of HRVB. (3) The physiological signals measured at the baseball field were blurred by noise such that subsequent signal analysis could not be effectively conducted. Therefore, the physiological results presented in this study comprise only laboratory measurements. (4) Grouping information was not hidden from the subjects, and so there may have been a placebo effect.

## 5. Conclusions

A 10-consecutive-day version of HRVB training was implemented in this study. The physiological results were consistent with previous studies in that participants in the experimental group showed an improvement in terms of regulatory capacity of the ANS. Notably, the test results indicated that through HRVB training, the batters had a better ability to reduce anxiety and improve their batting performance.

The results validate that the 10-consecutive-day version of HRVB is an effective intervention that can thus be applied before competition matches to help baseball players perform better. While these findings are promising, the limited sample size and power necessitate further research to confirm the generalizability of our results.

## Figures and Tables

**Figure 1 jfmk-10-00065-f001:**
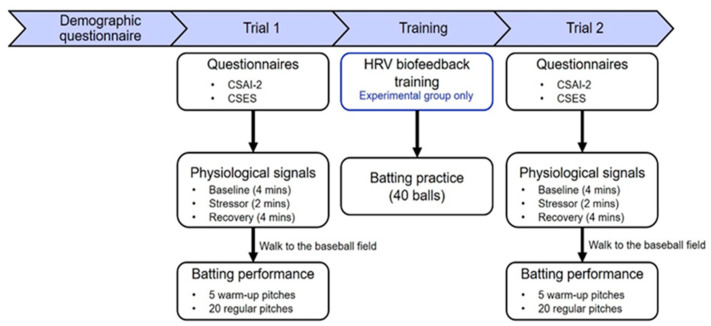
Experimental procedure used in this study. The blue areas represent different stages ordered chronologically.

**Figure 2 jfmk-10-00065-f002:**
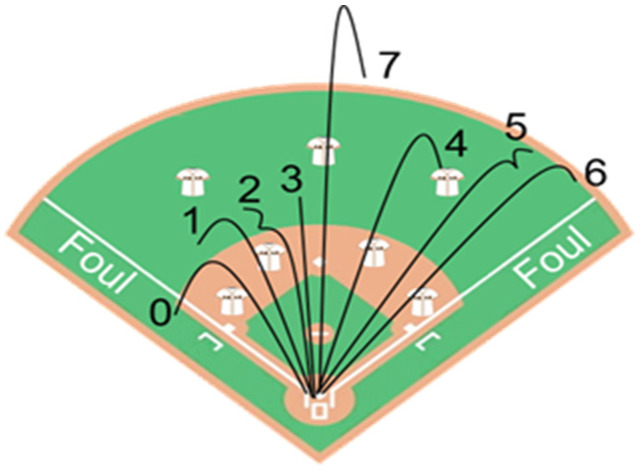
The point system proposed by Strack [7]. The numbers 1–7 correspond to scores.

**Figure 3 jfmk-10-00065-f003:**
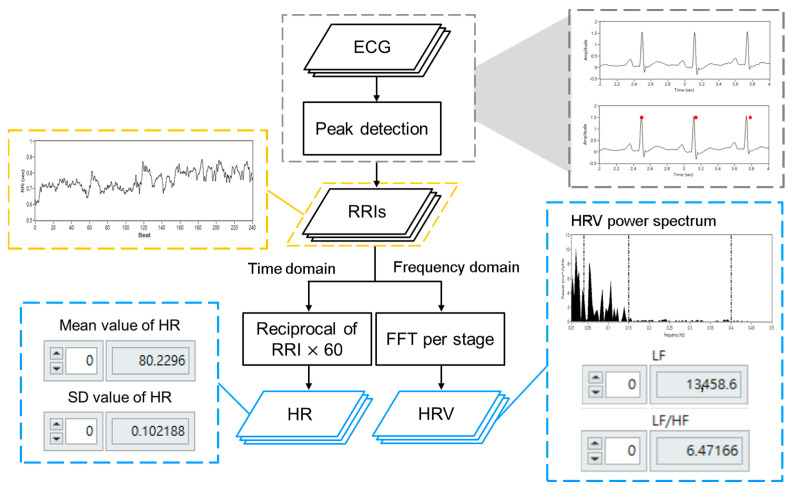
The processing flow of ECG signals.

**Figure 4 jfmk-10-00065-f004:**
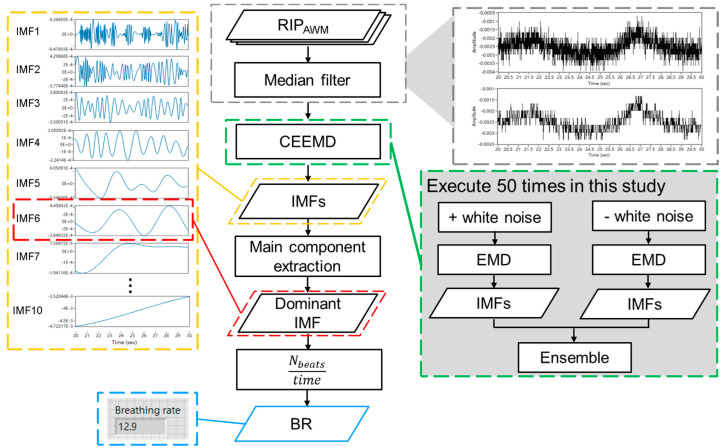
The processing flow of RIP signals.

**Figure 5 jfmk-10-00065-f005:**
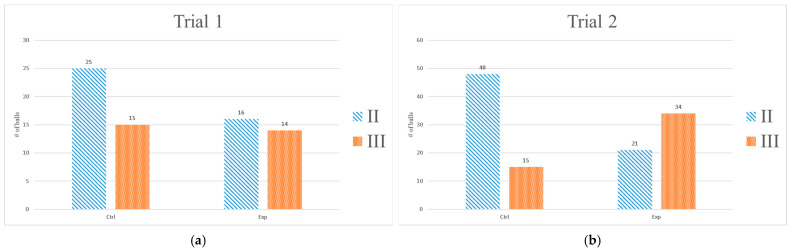
Number of balls in categories II and III in each group: (**a**) trial 1; (**b**) trial 2.

**Table 1 jfmk-10-00065-t001:** The demographic data of all participants.

Group (#)		Ctrl (9)	Exp (9)	Total (18)
Age (years)	μ±sd	21.2 ± 2.6	21.4 ± 2.0	20.9 ± 3.1
	Md	20.0	21.0	20.0
Playing baseball experiences (years)	μ±sd	6.4 ± 4.1	6.8 ± 4.4	6.0 ± 4.0
	Md	5.0	6.0	5.0
Height (cm)	μ±sd	175.9 ± 4.7	176.7 ± 4.5	175.1 ± 5.1
	Md	176.0	176.0	176.0
Weight (kg)	μ±sd	73.6 ± 9.9	71.1 ± 6.7	76.1 ± 12.3
	Md	70.0	68.0	72.0
BMI	μ±sd	23.8 ± 3.5	22.8 ± 1.7	24.9 ± 4.5
	Md	22.8	22.2	22.9

**Table 2 jfmk-10-00065-t002:** Descriptive statistics of HR and BR for all participants.

	Status	HR		BR	
		Ctrl	Exp	Ctrl	Exp
Trial 1	Baseline	86.9 ± 16.8	83.3 ± 14.2	18.9 ± 4.7	18.8 ± 5.7
	Stressor	90.1 ± 15.7	83.1 ± 14.6	24.5 ± 4.1	24.9 ± 5.6
	Recovery	86.5 ± 15.9	80.2 ± 14.1	18.9 ± 6.5	16.9 ± 6.3
Trial 2	Baseline	87.8 ± 9.5	80.3 ± 12.0	17.7 ± 4.2	14.6 ± 3.5
	Stressor	90.1 ± 8.1	83.1 ± 14.7	24.0 ± 3.2	19.8 ± 6.6 *
	Recovery	86.3 ± 8.9	78.4 ± 11.6	18.9 ± 3.8	17.9 ± 4.4

* *p* < 0.05 between the status in trial 1 and the corresponding status in trial 2.

**Table 3 jfmk-10-00065-t003:** Descriptive statistics for nLF, nHF, and LF/HF for all participants.

	Status	nLF		nHF		LF/HF	
		Ctrl	Exp	Ctrl	Exp	Ctrl	Exp
Trial 1	Baseline	76.7 ± 10.7	83.5 ± 9.8	23.3 ± 10.7	16.6 ± 9.8	4.1 ± 2.1	6.8 ± 4.5
	Stressor	75.0 ± 13.0	77.8 ± 14.5	25.0 ± 13.0	22.2 ± 14.5	4.2 ± 3.4	5.1 ± 3.1
	Recovery	82.2 ± 6.6	83.4 ± 12.3	17.8 ± 6.6	16.6 ± 12.3	5.4 ± 2.3	7.5 ± 4.1
Trial 2	Baseline	74.5 ± 9.0	88.9 ± 5.1 *^,‡^	25.5 ± 9.0	11.1 ± 5.1 *^,‡^	3.4 ± 1.5	9.5 ± 3.9 ^‡^
	Stressor	68.6 ± 12.4	80.2 ± 12.4 ^†^	31.4 ± 12.4	19.8 ± 12.4 ^†^	2.6 ± 1.2	6.8 ± 6.1 ^†^
	Recovery	75.9 ± 13.8	85.9 ± 9.7	24.1 ± 13.8	14.1 ± 9.7	5.1 ± 4.9	8.8 ± 5.2

* *p* < 0.05 between status in trial 1 and corresponding status in trial 2. ^†^  *p* < 0.05; ^‡^  *p* < 0.001 between control and experimental groups.

**Table 4 jfmk-10-00065-t004:** Variations in HR and BR values between trials.

Status	△HR		△BR	
	Ctrl	Exp	Ctrl	Exp
Baseline	0.9 ± 12.9	−3.1 ± 9.5	−1.1 ± 5.5	−4.2 ± 6.8
Stressor	−0.1 ± 13.6	0.0 ± 10.8	−0.4 ± 3.7	−5.1 ± 5.4
Recovery	−0.2 ± 12.8	−1.8 ± 12.3	0.0 ± 5.2	1.0 ± 7.7

**Table 5 jfmk-10-00065-t005:** Variations in nLF, nHF, and LF/HF values between trials.

Status	△nLF		△nHF		△LF/HF	
	Ctrl	Exp	Ctrl	Exp	Ctrl	Exp
Baseline	−2.3 ± 7.5	5.5 ± 6.8 ^†^	2.3 ± 7.5	−5.5 ± 6.8 ^†^	−0.7 ± 1.9	2.7 ± 5.2 ^†^
Stressor	−6.4 ± 8.7	2.5 ± 14.2 ^†^	6.4 ± 8.7	−2.5 ± 14.2 ^†^	−1.6 ± 2.8	1.7 ± 4.6
Recovery	−6.3 ± 10.7	2.5 ± 5.4 ^†^	6.3 ± 10.7	−2.5 ± 5.4 ^†^	−0.3 ± 4.8	1.2 ± 5.0

^†^  *p* < 0.05 between control and experimental groups.

**Table 6 jfmk-10-00065-t006:** Descriptive statistics for psychological parameters.

	Questionnaire	Ctrl	Exp
Trial 1	CSAI-2 (Cognitive anxiety)	23.67 ± 3.78	23.56 ± 4.07
	CSAI-2 (Somatic anxiety)	21.00 ± 5.57	19.44 ± 5.64
	CSAI-2 (Self-confidence)	21.89 ± 5.51	22.11 ± 5.33
	CSES	172.67 ± 23.61	168.11 ± 28.76
Trial 2	CSAI-2 (Cognitive anxiety)	24.00 ± 3.16	20.11 ± 4.78 *
	CSAI-2 (Somatic anxiety)	21.56 ± 6.02	17.22 ± 4.69
	CSAI-2 (Self-confidence)	21.89 ± 4.60	23.44 ± 3.47
	CSES	179.78 ± 23.11	167.44 ± 29.91

* *p* < 0.05 between the questionnaire in trial 1 and that in trial 2.

**Table 7 jfmk-10-00065-t007:** Descriptive statistics of batting performance.

	Batting Score		Infield Ratio (%)		Bat Speed (MPH)		Peak Hand Speed (MPH)	
	Ctrl	Exp	Ctrl	Exp	Ctrl	Exp	Ctrl	Exp
Trial 1	11.6 ± 11.7	9.8 ± 11.7	22 ± 22	17 ± 20	51.47 ± 4.64	56.56 ± 4.71	16.30 ± 1.47	17.74 ± 1.26
Trial 2	18.4 ± 11.8	19.8 ± 12.0 *	35 ± 23	31 ± 15 *	51.73 ± 5.49	57.07 ± 3.86 ^†^	16.76 ± 1.45	18.13 ± 0.98 ^†^

* *p* < 0.05 between trial 1 and trial 2. ^†^  *p* < 0.05 between control and experimental groups.

**Table 8 jfmk-10-00065-t008:** The number of balls in each category hit by each group in each trial.

	Trial 1		Trial 2	
Category	Ctrl	Exp	Ctrl	Exp
I	141	149	117	125
II	25	16	48	21
III	15	15	14	34

## Data Availability

The data presented in this study are available on request from the corresponding author (the data are not publicly available due to privacy or ethical restrictions).

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
