# Peer review of "The Impact of Heart Rate Variability Biofeedback on Anxiety Reduction and Batting Performance Enhancement in Taiwan University Baseball Players"

_jfmk, 2025, doi:10.3390/jfmk10010065_

Round 1
Reviewer 1 Report
Comments and Suggestions for Authors
Please find the comments.
The abstract should be structured. There are no materials and methods.
When was the study conducted?
Please indicate a gender category of the participants. This is missing.
Please indicate version of the SPSS.
Lines 166-168 must be deleted.
Effect size should be calculated for all the results.
The formatting of lines indicating statistical significance in the tables is messy. The results are presented in a vague way. Please indicate exact p-values with effect size for all the comparisons between the groups and within the groups at different time points. Please amend.
Presenting results in the discussion is a very uncommon practice. I feel there are serious methodological violations in this paper. It is prepared in a very specific uncommon way.
The sample size (n = 9) in each group is extremely small to detect differences at an appropriate level of power. The results seem to be questionable. Moreover, there are no descriptions of psychometric tests used in this study (i.e., CSAI-2, CSES). At least, internal consistency reliability should be calculated.
Reviewer 2 Report
Comments and Suggestions for Authors
Evaluation of manuscript jfmk-3385253
This manuscript evaluated the effect of heartrate variability on batting baseball performance. I will present below my considerations about the text.
Title: abbreviations should not be used in the title and it should be more specific. Ex: “Heart rate variability biofeedback improve the...
The Abstract needs numerical and statistical results, please review.
In the methods, how was sample representativeness calculated? How did the authors ensure equity of the groups before the experiment?
The tables should explain what the connectors are between two results.
The first paragraph of the discussion should restate the objective and summarize the main results.
The discussion is poor, the authors do not present explanations based on physiology regarding the association between HRV, anxiety and athletic performance in baseball.
Reviewer 3 Report
Comments and Suggestions for Authors
I would first like to commend the authors for their research. This is an interesting topic.
This paper does need significant revisions though for English grammar. I am going to highlight some below; however, before resubmitting I strongly recommend that someone whose primary language is English review the paper
L25: does not make sense; the use of mechanism is not common in this capacity
L27: coordinated
L29: Mental approach is not the common phrase used -or- how its grammar is presented is incorrect: Mental approach to hit is also (this is incomplete)
L32: "the emotion such as anxiety" is not proper English grammar
L25-L36: is all of this necessary. Would it be best just to focus on the mental side of things?
L38-L41: its a run on and does the 2nd part of the sentence mean anything?
L50: performance
L49-51: probably best as two sentences
L54-L68: this paragraph is hard to follow for 2 reasons: 1) the sentences describing each study are not either complete or there are grammar issues, 2) the extra words make it hard to follow, instead:
L58: Vachillo et al had golfers self-practice two times a day. They found golfers had significant improvements in HRV, LF, and HF.
spell out abbreviations at least one time prior to using abbreviations
L72: grammatically needs revision
L73-79: needs revision; remove excess/unnecessary words (work with someone who has English as primary language)
L86: spell out the exclusion criteria, don't use the word excluded twice in the same sentence
L95: lowercase competitive
L97: lowercase electrocardiogram, respiratory
L116: The training protocols were similar between the two groups; however, the experimental group received 20 minutes of HRVB per day for 10 days.
L119: we already know that they were recruited from the university, why does the season/semester info matter?
L128-136 need significant edits for punctuation
L157: delete in this study
What version of SPSS
Why were nonparametric stats used? No normality?
L166-168: is this text from the journal?
L170: Don't we know already that they are BB players? Delete
:184: have these been spelled out (abbrev)
Table 3: the lines are messy making it hard to read or understand what is going on, if you are trying to link the stat differences, you should put that below the table
Tables 5 and 7: have the same messy lines
Discussion, Table 8 and Figure 5, since this is data - it should go in the results, the discussion should put into context the results, not introduce new results
Comments on the Quality of English LanguageThis paper needs major revision for English grammar. It is very difficult to read at this time and I have suggested that they need to have someone with English grammar expertise to revise their paper
Round 2
Reviewer 1 Report
Comments and Suggestions for Authors
The paper has been improved, however, the main points have been not addressed.
The comment ("The sample size (n = 9) in each group is extremely small to detect differences at an appropriate level of power. The results seem to be questionable.") was not addressed.
Power analysis shows that the sample size is insufficient to address the research question (even the authors' power analysis showed a lack of appropriate power of 0.80 in most cases). The reliability of the results is questionable, conclusions could be misleading.
The authors argued that "Your concern regarding the sample size is valid. However, recruiting more participants proved challenging due to the COVID-19 pandemic (our experiment was conducted in 2020). To mitigate the potential impact of the small sample size, we employed non-parametric statistical methods. We believe that the sample size of this study is sufficient to demonstrate the utility of HRVB.", however, we have now 2025 year, so, there were a lot of opportunities to collect more data within all these years.
Another comment has not been addressed ("Moreover, there are no descriptions of psychometric tests used in this study(i.e., CSAI-2, CSES). At least, internal consistency reliability should be calculated."). As the tools were not described (e.g., number of items and subscales with their measured characteristics) and internal consistency reliability was not calculated, this also leads to serious doubts regarding the reliability of the obtained results. Readers should know what the psychological measures were applied and what the reliability (Cronbach's alpha and/or McDonald's omega) was. These are basic methodological requirements, which were presented with serious violations in this paper.
Several phrases are vague. For instance, "The statistical results of each questionnaire are shown in Table...". What do "statistical results" mean? "Statistical results of each questionnaire"? Questionnaires per se have no statistical results. They have scale or subscale scores which can be expressed in numbers using M, SD, Median etc.
Also, "Table 6. Statistical results for psychological parameters" presents numbers but it is unclear what these numbers mean. M +- SD? Or What? In the current form, the study is full of methodological problems, and it is unclear what and how was implemented in this paper.
Comments on the Quality of English LanguagePlease see above.
Reviewer 2 Report
Comments and Suggestions for Authors
The authors improve the quality of the manuscript. All my requests were met, therefore, I am in favor of acceptance.
Author Response
Thank you for your time and effort in reviewing our manuscript. We are glad to hear that all your requests have been addressed and that the quality of the manuscript has been improved. We appreciate your support for its acceptance
Reviewer 3 Report
Comments and Suggestions for Authors
I want to thank the authors, this is a much better presentation of your work and easier to follow now
I think you really addressed teh grammar related issues. I would like to see a final cleaned up version without all of the red changes. Might be nice for one final read through
Author Response
Thank you for your kind words and constructive feedback. We're glad to hear that the revised version is easier to follow and that the grammar issues have been addressed. We really appreciate your time and effort in reviewing our manuscript.
Round 3
Reviewer 1 Report
Comments and Suggestions for Authors
The authors provided several explanations regarding methodological limitations. I would prefer to leave the assessment of the relevance of these explanations to the editors.
It is unclear how was the effect size of the differences of scores between groups (based on M-U test) calculated using G*Power? First of all, as an effect size measure, M-U test uses rank-biserial correlation which can be from -1 to 1. However, in some cases, the authors stated effect sizes higher that [-1; +1]. For instance, "Moreover, in the baseline and stressor phases of trial 2, significant differences were found in the nLF (p 328 = 0.002 and 0.024; power = 0.97 and 0.44; effect size = 2.68 and 0.94, respectively) and LF/HF 329 (p = 0.002 and 0.024; power = 0.98 and 0.46; effect size = 2.78 and 0.97, respectively) between the two groups."
What is the measure of this effect size which is higher that I1I? It is definitely not rank-biserial correlation coefficient which should be used instead. The same issue is applied to Wilcoxon-Test, which uses r effect size, and other elements which should be clarified.
Téllez A., García C.H., Corral-Verdugo V. (2015). Effect size, confidence intervals and statistical power in psychological research. Psychology in Russia: State of the Art, 8(3), 27-47.
As I previously mentioned, the methodological descriptions in this paper are somewhat vague.
